

# Necessity of electrically conductive pili for methanogenesis with magnetite stimulation

Oumei Wang[1], Shiling Zheng[2], Bingchen Wang[2,3], Wenjing Wang[2] and Fanghua Liu[2]

[1] Binzhou Medical University, Yantai, Shandong, China
[2] Key Laboratory of Coastal Biology and Biological Resources Utilization, Yantai Institute of Coastal Zone Research, Chinese Academy of Sciences, Yantai, Shandong, China
[3] University of Chinese Academy of Sciences, Beijing, China

## ABSTRACT

**Background**. Magnetite-mediated direct interspecies electron transfer (DIET) between *Geobacter* and *Methanosarcina* species is increasingly being invoked to explain magnetite stimulation of methane production in anaerobic soils and sediments. Although magnetite-mediated DIET has been documented in defined co-cultures reducing fumarate or nitrate as the electron acceptor, the effects of magnetite have only been inferred in methanogenic systems.

**Methods**. Concentrations of methane and organic acid were analysed with a gas chromatograph and high-performance liquid chromatography, respectively. The concentration of HCl-extractable Fe(II) was determined by the ferrozine method. The association of the defined co-cultures of *G. metallireducens* and *M. barkeri* with magnetite was observed with transmission electron micrographs.

**Results**. Magnetite stimulated ethanol metabolism and methane production in defined co-cultures of *G. metallireducens* and *M. barkeri*; however, magnetite did not promote methane production in co-cultures initiated with a culture of *G. metallireducens* that could not produce electrically conductive pili (e-pili), unlike the conductive carbon materials that facilitate DIET in the absence of e-pili. Transmission electron microscopy revealed that *G. metallireducens* and *M. barkeri* were closely associated when magnetite was present, as previously observed in *G. metallireducens/G. sulfurreducens* co-cultures. These results show that magnetite can promote DIET between *Geobacter* and *Methanosarcina* species, but not as a substitute for e-pili, and probably functions to facilitate electron transfer from the e-pili to *Methanosarcina*.

**Conclusion**. In summary, the e-pili are necessary for the stimulation of not only *G. metallireducens/G. sulfurreducens*, but also methanogenic *G. metallireducens/M. barkeri co-cultures* with magnetite.

Corresponding authors
Oumei Wang, omwang@aliyun.com
Fanghua Liu, fhliu@yic.ac.cn

## INTRODUCTION

Microbial methane production is one of the most successful, large-scale bioenergy strategies (*Liu et al., 2009*; *Shen et al., 2016*) and methane production in terrestrial environments is
a major source of atmospheric methane, an important greenhouse gas (*Bridgham et al., 2013*; *Conrad, 2007*). In freshwater methanogenic environments, and anaerobic digesters, methanogens primarily produce methane from the metabolism of acetate and the reduction of carbon dioxide with $H_2$ to methane. The well-known source of electrons for carbon dioxide reduction to methane is $H_2$ (*Sieber, McInerney & Gunsalus, 2012*); however, it has recently been demonstrated that *Methanosaeta* and *Methanosarcina* species can accept electrons from the donor strain *G. metallireducens* for carbon dioxide reduction via direct interspecies electron transfer (DIET) (*Chen et al., 2014a*; *Chen et al., 2014b*; *Rotaru et al., 2014a*; *Rotaru et al., 2014b*; *Wang et al., 2016*).

In the absence of added conductive materials, DIET between *Geobacter metallireducens* and *Methanosaeta* and *Methanosarcina* species requires the electrically conductive pili (e-pili) of *G. metallireducens* (*Chen et al., 2014a*; *Rotaru et al., 2014a*; *Rotaru et al., 2014b*). The e-pili of both *Geobacter* species are also required for DIET in co-cultures of *G. metallireducens* and *G. sulfurreducens* (*Shrestha et al., 2009*; *Summers et al., 2010*). Existing studies on the e-pili of *G. sulfurreducens* have suggested that the conductivity along the length of *Geobacter* e-pili (*Adhikari et al., 2016*; *Malvankar & Lovley, 2014*) can be attributed to the tight packing of aromatic amino acids within the pilus structure, which confer a metallic-like conductivity similar to that observed in carbon nanotubes (*Malvankar et al., 2015*; *Malvankar et al., 2011*; *Vargas et al., 2013*). The e-pili are decorated with the *c*-type cytochrome OmcS, which does not contribute to conductivity along the length of the e-pili, but is important for electron transfer from the e-pili to extracellular electron acceptors/donors (*Leang et al., 2010*; *Liu et al., 2015*; *Malvankar & Lovley, 2014*; *Malvankar, Tuominen & Lovley, 2012*; *Mehta et al., 2005*; *Summers et al., 2010*). It is expected that the e-pili of *G. metallireducens* function in a similar manner (*Smith, Lovley & Tremblay, 2013*; *Tremblay et al., 2012*; *Zheng et al., 2017*), but the cytochrome(s) that are attached to the e-pili of *G. metallireducens* have not yet been identified.

Conductive carbon materials, such as: granular activated carbon, carbon cloth, and biochar, stimulate DIET (*Chen et al., 2014a*; *Chen et al., 2014b*; *Liu et al., 2012*; *Rotaru et al., 2014a*). The electron-donating and electron-accepting partners both attach to the conductive carbon materials, which serve as an electrical conduit between the two species. Mutant *Geobacter* strains that lack e-pili can participate in DIET under these conditions, presumably because biological cell-to-cell electrical conduits are no longer required (*Chen et al., 2014a*; *Chen et al., 2014b*; *Liu et al., 2012*; *Rotaru et al., 2014a*).

An important insight into carbon and electron flow in methanogenic environments lies in the finding that magnetite stimulated methane production in enrichment cultures inoculated from paddy soil with either ethanol or acetate as the electron donor (*Kato, Hashimoto & Watanabe, 2012a*). The enhanced methane production was accompanied by an enrichment of the *Geobacter* and *Methanosarcina* species (*Kato, Hashimoto & Watanabe, 2012a*). It was hypothesised that the magnetite provided electrical contact between the *Geobacter* and *Methanosarcina* species and that the *Geobacter* species oxidized the ethanol or acetate to carbon dioxide with electron transfer to the *Methanosarcina*, which then used the electrons to reduce carbon dioxide to methane (*Kato, Hashimoto & Watanabe, 2012a*). Many subsequent studies have documented the fact that magnetite

accelerates methane production in samples from sediments or anaerobic digesters or defined co-cultures and have also inferred that this can be attributed to enhanced electron transfer through magnetite to methanogens (*Li et al., 2015*; *Tang et al., 2016*; *Yang et al., 2015*; *Zhuang et al., 2015*). Magnetite does promote interspecies electron transfer between *Geobacter sulfurreducens* and *Thiobacillus denitrificans* growing with acetate as the electron donor and nitrate as the electron acceptor (*Kato, Hashimoto & Watanabe, 2012b*), as well as between *G. metallirducens* and *G. sulfurreducens* growing with ethanol as the electron donor and fumarate as the electron acceptor (*Liu et al., 2015*), however, it has never been directly demonstrated that magnetite promotes DIET to methanogens. Analysis of the mechanisms by which magnetite enhanced DIET in *G. metallireducens/G. sulfurreducens* co-cultures indicated that, unlike conductive carbon materials, magnetite does not act as a substitute for e-pili, but rather can take the place of OmcS by attaching to e-pili to facilitate DIET, thus alleviating the need for OmcS production (*Liu et al., 2015*). Therefore, it should not be assumed that magnetite promotes DIET to methanogens as has been demonstrated for conductive carbon materials. The purpose of this study was to evaluate further the possibility that magnetite promotes DIET to methanogens.

## MATERIALS AND METHODS

### Microorganisms, media, and growth conditions

Wild-type *Geobacter metallireducens* strain GS-15 (ATCC 53774) (*Aklujkar et al., 2009*; *Lovley et al., 1993*) and a strain of *G. metallireducens* in which the gene for PilA, the pilus monomer, was deleted (*Tremblay et al., 2012*) were obtained from our laboratory collection. *Methanosarcina barkeri* strain DSM 800 (ATCC 43569) was obtained from DSMZ (Braunschweig, Germany).

All culturing was performed under strict anaerobic conditions under a gas phase of $N_2/CO_2$ (80/20). Inocula for co-cultures were developed by growing *G. metallireducens* strains in Fe(III)-citrate (FC) medium (*Bagnara et al., 1985*), with 20 mmol $L^{-1}$ ethanol as the sole electron donor and 55 mmol $L^{-1}$ Fe(III) citrate as the electron acceptor. For co-cultures of *G. metallireducens* and *M. barkeri*, *G. metallireducens* was grown in DSMZ methanogenic medium 120 with 20 mmol $L^{-1}$ ethanol as the electron donor and nitrate (10 mmol $L^{-1}$) as the electron acceptor. *M. barkeri* was grown in the same medium with 30 mmol $L^{-1}$ acetate as the substrate. Co-cultures were grown in 40 mL medium 120 with a 10% inoculum and with ethanol (20 mmol $L^{-1}$) as the electron donor as described previously (*Rotaru et al., 2014a*). The incubation temperature for all methanogenic studies was 37 °C. When noted, magnetite was prepared as previously described (*Kang et al., 1996*) and added from stock solutions to give a final concentration of 5 mmol $L^{-1}$ before autoclaving.

### Chemical analysis

The gaseous samples were regularly collected from enrichment cultures with pressure-lock analytical syringes. The concentrations of $CH_4$ were analysed with a gas chromatograph (GC-7890A; Agilent Technologies, Santa Clara, CA, USA) equipped with a flame ionisation detector.

Concentrations of ethanol and acetate were analysed with high-performance liquid chromatography (1260 Infinity; Agilent Technologies, Santa Clara, CA, USA) with a Hi-plex H column equipped with a refractive index detector.

The concentration of HCl-extractable Fe(II) was extracted from cultures and each replicate of the assays in triplicate as described previously (*Zheng et al., 2015*). Moreover, the concentration of dissolved Fe(II) in samples was also quantified by filtering through 0.45 μm sterile syringe filters and using the ferrozine method as described above.

### Microscopy

Samples of cells and magnetite were negatively stained with 2% phosphotungstic acid and examined by a JEM-1400 (JEOL, Japan) transmission electron microscope (TEM).

## RESULTS AND DISCUSSION

### Magnetite stimulation of DIET between *G. metallireducens* and *M. barkeri*

To evaluate whether, or not, magnetite was capable of stimulating DIET between *G. metallireducens* and *M. barkeri*, co-cultures were initiated with ethanol as the sole electron donor in the presence, and absence, of magnetite. Although *M. bakeri* is capable of using $H_2$ as an electron donor, *G. metallireducens* cannot metabolise ethanol with the production of $H_2$ (*Rotaru et al., 2014b*; *Shrestha et al., 2013a*; *Summers et al., 2010*) and thus syntrophic growth in *G. metallireducens/M. barkeri* co-cultures can be attributed to DIET. First, based on Fig. 1A, the production of $CH_4$ from the co-culture of *G. metallireducens* and *M. barkeri* without magnetite indeed indicates DIET in syntrophic interaction. However, DIET is not exclusively occurred, at least a part of methane was still produced from acetate. For example, if half of the methane (∼0.25 mmol) was produced from acetate (∼0.25 mmol), the concentration of acetate remained in the medium should be ∼0.25 mmol.

The initial establishment of *G. metallireducens* and *M. barkeri* co-cultures requires a long adaption period in the absence of added conductive materials (*Rotaru et al., 2014a*). As expected, ethanol was only slowly metabolised over 50 days without magnetite (Fig. 1C), however, in the presence of magnetite, ethanol was metabolised with the production of methane beginning within 10 days (Fig. 1C). Non-inoculated controls with magnetite showed no ethanol metabolism or methane production.

Limited acetate accumulated in the *G. metallireducens* with, or without, *M. barkeri* co-cultures in the presence of magnetite ($C_2H_6O + H_2O \rightarrow C_2H_4O_2 + 4H^+ + 4e^-$, Oxidation of one ethanol will produce one acetate plus four electrons released (*Rotaru et al., 2014a*)), but was later consumed (Fig. 1D), which differed from co-cultures of *G. metallireducens* and *M. barkeri* in the absence of magnetite, suggesting that *G. metallireducens* metabolised the acetate that *G. metallireducens* produced from ethanol compared with the result of *G. metallireducens* acting alone with magnetite. The high amount of methane in the *G. metallireducens* and *M. barkeri* co-cultures suggested that *M. barkeri* only used the electrons released from ethanol oxidation for reducing carbon dioxide to produce methane in the magnetite-amended co-cultures ($8H^+ + 8e^- + CO_2 \rightarrow CH_4 + 2H_2O$). The total amount of ethanol from the magnetite-amended co-cultures metabolised

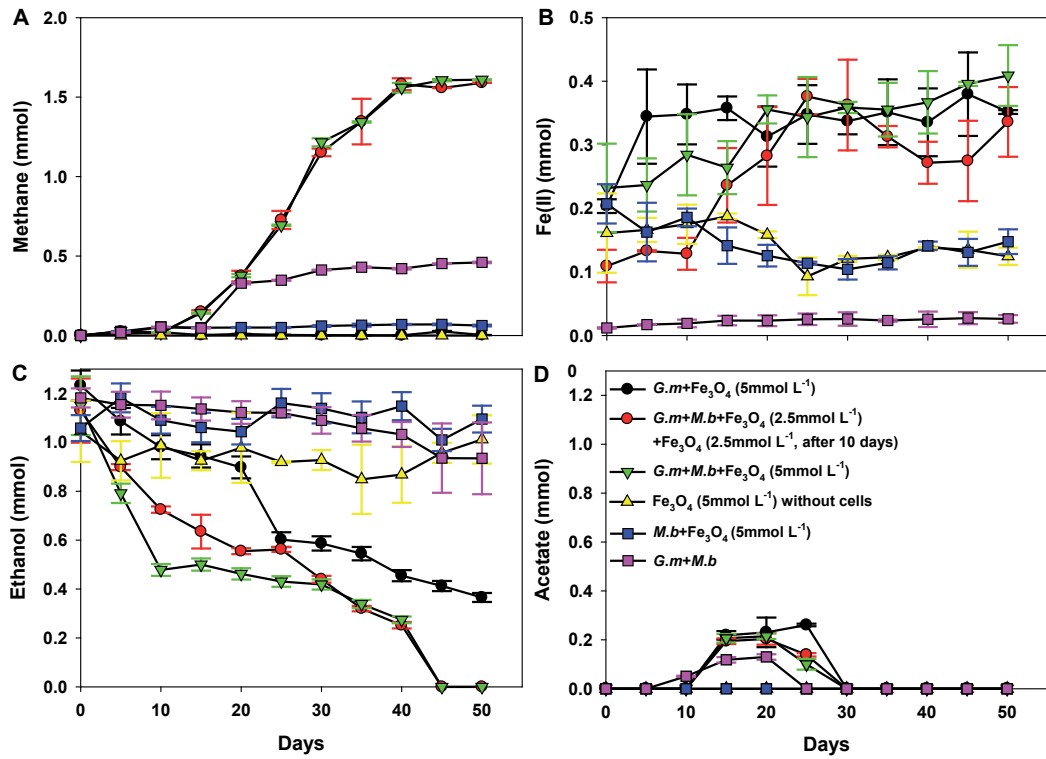

**Figure 1** **Co-cultures of *G. metallireducens* (*G. m*) and *M. barkeri* (*M. b*) with ethanol as the substrate in the presence, or absence, of magnetite (Fe₃O₄).** Quantities of methane (A), ferrous iron (B), ethanol (C), and acetate (D) in cultures. Data are the means and standard deviation for triplicate cultures. In some instances the standard deviation was less than the size of the symbol.

$(1.15 \pm 0.12$ mmol) resulted in $1.60 \pm 0.0032$ mmol methane (Figs. 1A, 1C), which showed that the mmol ratio of $CH_4/C_2H_6O$ (1.60/1.15) was 1.39 (>1), thus about 92.2% of the electrons from ethanol oxidation were recovered in methane according to the equation: $2C_2H_5OH \rightarrow 3CH_4 + CO_2$. Furthermore, no $H_2$ was detected in any of the experiment groups. This result was consistent with the fact that *G. metallireducens* is unable to produce $H_2$ during ethanol metabolism (*Shrestha et al., 2013b*). Therefore, the high electron recovery that was available from ethanol to methane suggested that magnetite can stimulate DIET between *G. metallireducens* and *M. barkeri* and suggested that the simplest explanation for the enrichment of *Geobacter* and *Methanosarcina* observed in the presence of magnetite in previous studies (*Kato, Hashimoto & Watanabe, 2012a*) is that magnetite was facilitating DIET.

HCl-extractable ferrous iron was also produced in *G. metallireducens-M. barkeri* co-cultures from reduction ferric iron of magnetite within five days and increased to $0.1768 \pm 0.0219$ mmol at 50 days, which was equal to that when *G. metallireducens* was tested with magnetite alone ($0.1761 \pm 0.0549$ mmol) (Fig. 1B); however, the concentration of dissolved ferrous iron was under detect limitation during the incubation of the co-cultures amended with magnetite, suggested that only a part of the ferric iron in the added

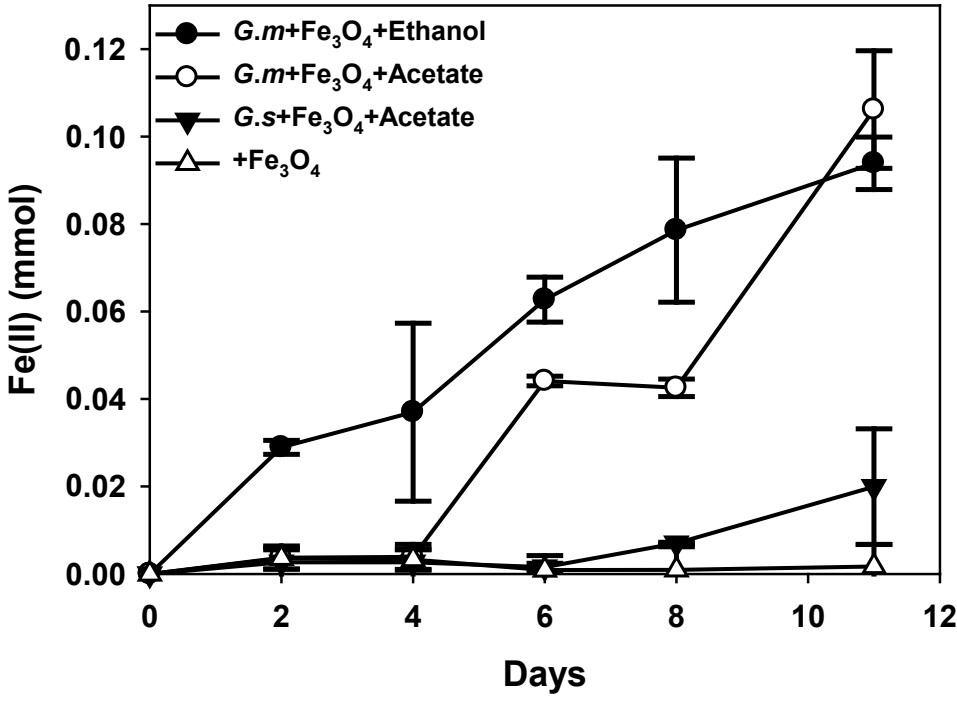

**Figure 2** Quantities of ferrous iron in cultures of *Geobacter metallireducens* (*G.m*) and *Geobacter sulfurreducens* (*G.s*) in the presence of magnetite with ethanol and acetate as the substrates.

magnetite was reduced to ferrous iron. The results indicated that only a small portion of electrons (about 4.6 mmol electrons released from 1.15 mmol ethanol oxidation, 0.18 mmol/4.6 mmol, about 3.9%) in *G. metallireducens-M. barkeri* co-cultures with magnetite were used for ferric iron reduction and the majority of electrons (about 96.1%) were used for methane production. This result differs from that reporting that magnetite acts as the electrical conduit between electron-donating *Geobacter* and electron-accepting methanogens (*Kato, Hashimoto & Watanabe, 2012a*; *Li et al., 2015*; *Viggi et al., 2014*). One factor controlling ferrous iron production in co-cultures amended with magnetite is the range of substrates that can be metabolised by *Geobacter* species. *G. metallireducens* can utilise ethanol and acetate, ferrous iron production from acetate was slower than that from ethanol within 10 days in the presence of magnetite, while ferrous iron production from *G. sulfurreducens* amended with magnetite was much lower than that from *G. metallireducens* when utilising acetate (Fig. 2). This suggested that *G. metallireducens*, like some microorganisms (e.g., *Shewanella*, *Dechloromonas*, *Desulfovibrio*, and *Clostridium*) was able to use magnetite as the electron acceptor from ethanol or acetate metabolism (*Kostka & Nealson, 1995*; *Yang et al., 2015*). However, it is not possible for magnetite to act as the electron shuttle for production of methane from carbon dioxide because of the relatively high mid-point potential of the Fe(III)/Fe(II) redox couple ($E_0$' = +0.20 V, pH 7.0) which is too high for the reduction of carbon dioxide to methane ($E_0$' of $CO_2$/methane couple = −240 mV).

To determine the actual role of magnetite in stimulation of ethanol metabolism and methane production in co-cultures of wild-type *G. metallireducens* and *M. barkeri*, co-cultures were initiated with 2.5 mmol $L^{-1}$ magnetite, after a 10-day incubation period, an additional 2.5 mmol $L^{-1}$ magnetite was subsequently added. Methane production presented the same tendency with 5 mmol $L^{-1}$ magnetite added to the co-cultures (Fig. 1A): this meant that the manner and amount of addition of magnetite could not affect methane production, however, the amount of HCl-extractable ferrous iron changed: the reduced ferrous iron concentration was about 0.0193–0.0239 mmol (∼9.6–12% of added $Fe^{3+}$) when 2.5 mmol $L^{-1}$ magnetite ($Fe^{3+}$: 0.2 mmol) was added during the first 10 days, and subsequently reduced when more magnetite was added, the amount of ferrous iron used in each step (total: $0.1635 \pm 0.0313$ mmol) was similar to the addition of 5 mmol $L^{-1}$ magnetite (Fig. 1B). This result was consistent with the observation that *G. metallireducens* alone reduced magnetite to produce ferrous iron (Fig. 1B). Thus, the initial concentration of magnetite determined how much Fe(III) inside was reduced. When high concentration of magnetite (5 mmol $L^{-1}$) was available, Fe(III) reduction was detected; however, no significant Fe(III) reduction was found when lower concentration of magnetite (2.5 mmol $L^{-1}$) was present. Fe(III) in the magnetite was reduced only when additional magnetite (2.5 mmol $L^{-1}$) was added. This demonstrated that lower concentration of magnetite could not be preferentially used as the electron acceptor in the co-culture of *G. metallireducens* and *M. barkeri*. Similarly, ethanol was stimulated to oxidise and little acetate was transiently accumulated in magnetite upon its step-by-step addition to co-cultures of wild-type *G. metallireducens* and *M. barkeri* (Figs. 1C, 1D). The calculation of electron recovery (93.81%) of electrons available from ethanol in methane in these samples further suggested that *M. barkeri* was accepting electrons from carbon dioxide reduction via DIET.

Transmission electron microscopy (TEM) revealed that *G. metallireducens* (rod-shaped cells) and *M. barkeri* (larger size cocci) were associated with each other (Fig. 3A). With higher magnification it was apparent that magnetite was associated with the *G. metallireducens* pili (Fig. 3B), as was previously observed that some of the magnetite was localised along pili and compensated for the lack of OmcS of *G. sulfurreducens* in promoting electrical contacts with pili in *G. metallireducens/G. sulfurreducens* co-cultures (*Liu et al., 2015*).

## Failure of magnetite to compensate for loss of e-pili in *G. metallireducens*

To investigate further the mechanisms for magnetite stimulation of DIET between *G. metallireducens* and *M. barkeri*, co-cultures were initiated with the previously described strain of *G. metallireducens* (*Tremblay et al., 2012*) that is incapable of producing pili because the gene for PilA, the pilus monomer, has been deleted. As expected from previous studies (*Rotaru et al., 2014a*), methane was not produced in co-cultures with the pili-deficient strain of *G. metallireducens* (Fig. 4A); however, co-cultures amended with magnetite produced less methane (about $0.38 \pm 0.025$ mmol, Fig. 4A). During co-culture testing, ferrous iron concentrations were below 0.1 mmol in magnetite amended

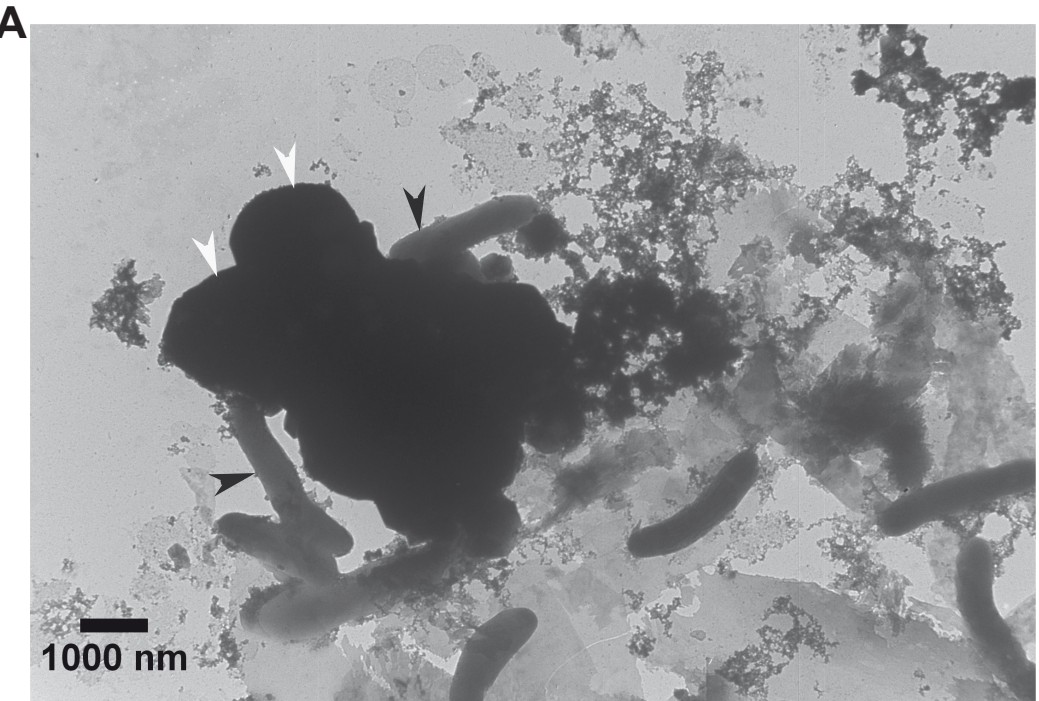

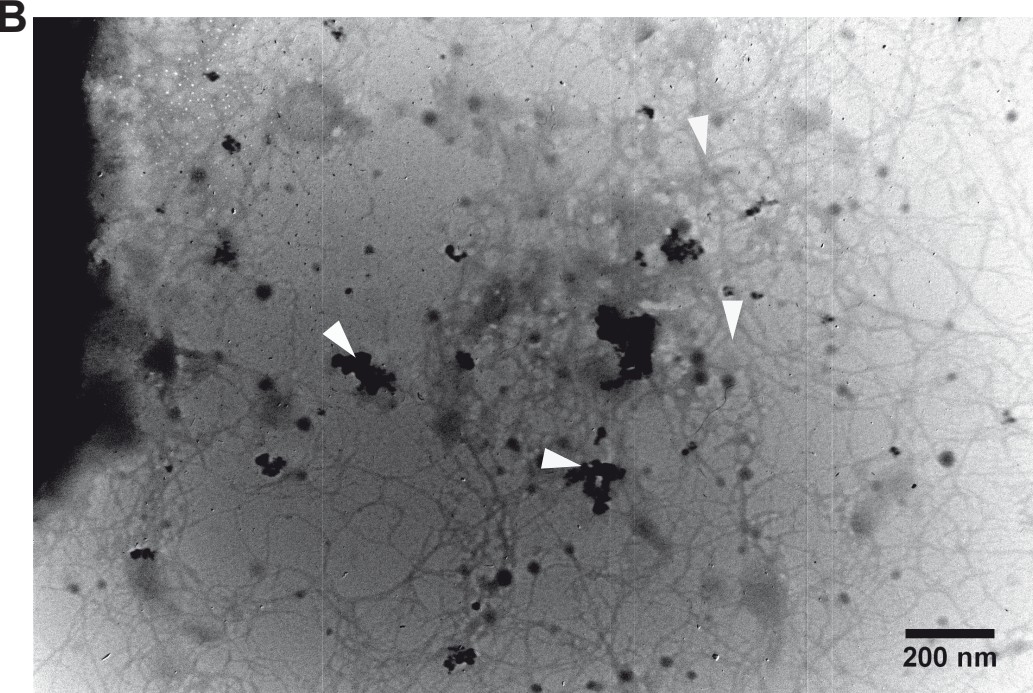

**Figure 3** **Transmission electron micrographs.** Association of the defined co-cultures of *G. metallireducens* and *M. barkeri* with magnetite. (A) Association of the two cell types. Black and white arrows indicate *G. metallireducens* cells and *M. barkeri* cells, respectively. (B) Association of magnetite with pili. White arrows indicate magnetite and pili.

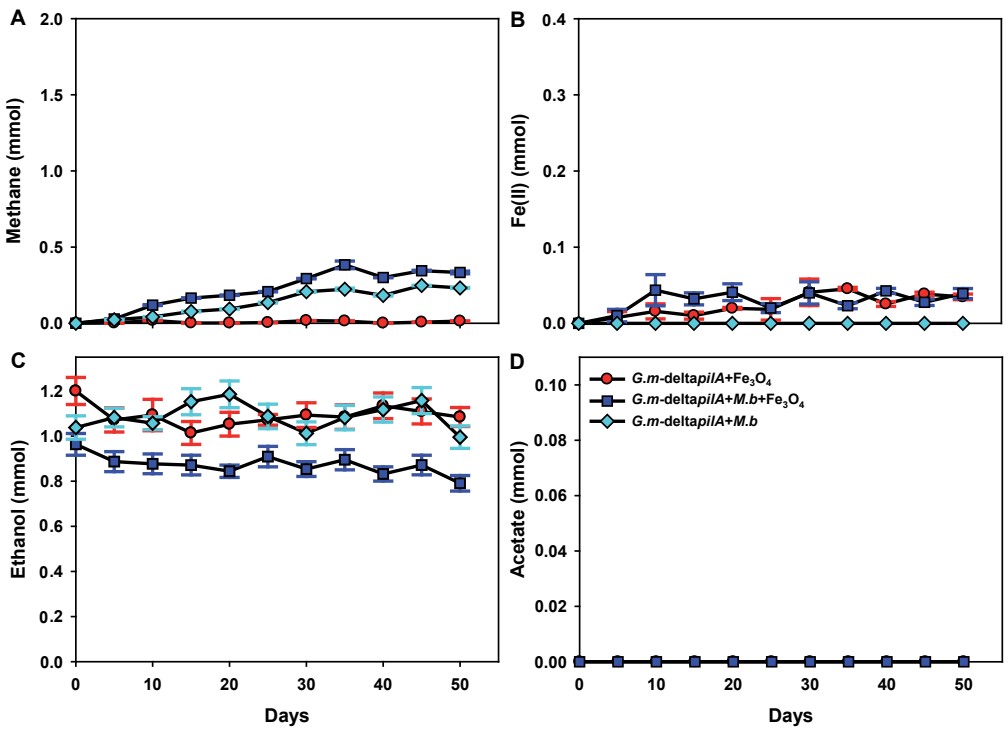

**Figure 4** **Co-cultures of *M. barkeri* (*M. b*) and a *PilA-deficient G.metallireducens* (*G. m-deltapilA*) strain in the presence, or absence, of magnetite (Fe₃O₄).** Quantities of methane (A), ferrous iron (B), ethanol (C), and acetate (D) in cultures. Data are the means and standard deviation for triplicate cultures. In some instances the standard deviation was less than the size of the symbol.

cultures (Fig. 4B). Furthermore, co-cultures with the *pil* A-deficient *G. metallireducens* failed to metabolise ethanol or produce acetate with, or without, magnetite amendment (Figs. 4C, 4D). These results suggested that magnetite perhaps can partly substitute for pili to participate in DIET of co-cultures resulting from its electrical conductivity; however, magnetite appears to promote DIET by a mechanism that is different than that in conductive carbon materials such as GAC and carbon cloth (*Chen et al., 2014a*; *Liu et al., 2012*). In the presence of GAC or carbon cloth the pili-deficient strain of *G. metallireducens* can transfer electrons to *M. barkeri* because both species attach to the conductive materials, which are much bigger than individual cells. Magnetite particles are typically smaller (at 20–50 nm) than cells and thus are unlikely to provide effective cell-to-cell contacts (*Liu et al., 2015*). This was evident in previous studies with *G. metallireducens/G. sulfurreducens* co-cultures in which magnetite was not able to compensate for the lack of e-pili in *G. metallireducens* (*Liu et al., 2015*). Multiple lines of evidence, including studies with an OmcS-deficient mutant, suggested that magnetite could serve as the functional equivalent of OmcS, and the *c*-type cytochrome associated with the e-pili of *G. sulfurreducens* (*Liu et al., 2015*). Similar genetic experiments are not yet possible with *G. metallireducens* because the cytochrome(s) associated with the *G. meatllireducens* e-pili have not been identified. However, the finding that magnetite amendments did not permit the growth of

the pili-deficient strain of *G. metallireducens* in co-culture with *M. barkeri*, suggested the magnetite cannot function as an e-pili substitute in all regards. Magnetite was associated with the e-pili in the *G. metallireducens/M. barkeri* co-cultures. Therefore, it is likely that magnetite also facilitated electron transfer from the *G. metallireducens* e-pili to *M. barkeri* in the co-cultures.

## CONCLUSIONS

In sum, we have established co-cultures of *M. barkeri* and wild-type *G. metallireducens* or a strain deficient in the PilA gene with or without magnetite. The results revealed magnetite stimulated ethanol metabolism and methane production in defined co-cultures of *G. metallireducens* and *M. barkeri*. However, magnetite did not promote methane production in co-cultures of the *pilA*-deficient *G. metallireducens*. These results showed that magnetite could not substitute for e-pili to promote DIET between *Geobacter* and *Methanosarcina* species, in which the e-pili are necessary for the stimulation.

### Funding

This research was supported by the Major Research Plan (No. 91751112) and the General Programme (No. 41573071 and 41371257) of the National Natural Science Foundation of China, the Key Research Project of Frontier Science (No. QYZDJ-SSW-DQC015) of the Chinese Academy of Sciences, the Natural Science Foundation of Shandong Province (Grant no. ZR2016DQ12), and the Young Taishan Scholars Programme (No. tsqn20161054). The funders had no role in study design, data collection and analysis, decision to publish, or preparation of the manuscript.

### Grant Disclosures

The following grant information was disclosed by the authors:
Major Research Plan: 91751112.
National Natural Science Foundation of China: 41573071, 41371257.
Chinese Academy of Sciences: QYZDJ-SSW-DQC015.
Natural Science Foundation of Shandong Province: ZR2016DQ12.
Young Taishan Scholars Programme: tsqn20161054.

### Competing Interests

The authors declare there are no competing interests.

### Author Contributions

- Oumei Wang conceived and designed the experiments, approved the final draft.
- Shiling Zheng performed the experiments, analyzed the data, prepared figures and/or tables, approved the final draft.
- Bingchen Wang performed the experiments, contributed reagents/materials/analysis tools, approved the final draft.

- Wenjing Wang performed the experiments, approved the final draft.
- Fanghua Liu conceived and designed the experiments, authored or reviewed drafts of the paper, approved the final draft.

## Data Availability

The raw data have been uploaded as Supplemental Files.

## Supplemental Information

Supplemental information for this article can be found online at http://dx.doi.org/10.7717/peerj.4541#supplemental-information.

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
