# Peer review of "Necessity of electrically conductive pili for methanogenesis with magnetite stimulation"

_PeerJ, doi:10.7717/peerj.4541_

## Round 0.1 · original submission · Major Revisions

According to the comments raised by the two reviewers, this manuscript requires Major Revisions before consideration of acceptance for publication. The comments from Reviewer #2 are very critical and important to the conclusion of the manuscript. Please carefully address all the issues and refine the wording to describe the results and discussion more precisely.

Reviewer 1 ·

Basic reporting

no comment

Experimental design

no comment

Validity of the findings

1. I am not convinced that G. metallireducens and M. barkeri were closely associated by TEM in Fig. 3A. It is too dark to figure out what are they. This is important to the conclusion.

2. In the co-cultures of G. metallireducens and M. barkeri with ethanol as the substrate in the absence of magnetite, the dynamic concentrations of methane, ethanol and acetate are not well matched (pink lines in Fig.1).

3. line 188: magnetite was firstly reduced by G. metallireducens... Check your data in Fig.1, the changes in Fe(II) concentrations are insignificant during the first 10 days.

Additional comments

Using the co-cultures of M. barkeri and wild-type G. metallireducens or a PilA-deficient strain, the authors demonstrated that the presence of magnetite accelerated the conversion of ethanol to methane in defined co-cultures of G.metallireducens and M. barkeri. More importantly, they found that magnetite could not substitute for e-pili to promote DIET between Geobacter and Methanosarcina. This is a fair contribution to DIET research. Paper is well written, the enhanced quality of figure is need to support results, and the data interpretation should be more rigorous.

·

Basic reporting

I read the manuscript without any language problem. I found the literature review was complete and references were properly cited. The layout of the presentation including main text and figs is acceptable.

Experimental design

The work represents a piece of original primary research. The question, whether magnetite can substitute e-pili for direct interspecies electron transfer (DIET), was well defined. To answer this question, the defined cocultures were established using the wild type Geobacter metallireducens that produce e-pili or its mutant that can not produce e-pili with Methanosarcina barkeri. These cocultures were used to test the effect of magnetite.

Validity of the findings

My major concern is the reduction of Fe(III) in magnetite by G. metallireducens. It was clear that G. metallireducens can use both ethanol and acetate to reduce Fe(III) from magnetite (Fig. 1C and Fig. 2). When the wild type coculture was tested with ethanol, acetate was produced and used by methanogen. Oxidation of one ethanol will produce one acetate plus two electrons released. The stoichiometry (line 155) and data in Fig. 1D indicated that the acetate produced was completely dismutated into CH4 and CO2. This reaction was completed by M. barkeri alone and should be nothing to do with DIET. Apparently, the addition of magnetite promoted the activity of G. metallireducens by providing a proper electron acceptor. Without clarifying this fact, the statement that magnetite promoted DIET (from line 157 to 161) can be misleading.
The test on mutant coculture suggests that the e-pili is essential for the reduction of (Fe(III) from magnetite. Thus, reduction activity appears dominating the conversion from ethanol to acetate and then to CH4 and CO2 in the wild type coculture. The reduction activity makes the explanation complicated and eventually can end up a completely different story from DIET. The statement about the effect of magnetite on DIET can be valid only if this reduction activity can be quantitatively clarified or excluded.

---

## Round 0.2 · Minor Revisions

Please carefully address the two comments raised by one of the reviewers and provide necessary discussion on them in the maint text of the revised manuscript.

Reviewer 1 ·

Basic reporting

My comments have been addressed properly.

Experimental design

My comments have been addressed properly.

Validity of the findings

My comments have been addressed properly.

Additional comments

My comments have been addressed properly.

·

Basic reporting

no comment

Experimental design

no comment

Validity of the findings

no comment

Additional comments

The manuscript is greatly improved. But some data still look odd to me and I shall appreciate if authors can give more explanations. First, based on Figure 1A, the production of CH4 from the coculture of G. metallireducens and M. barkeri without magnetite indeed indicates DIET in syntrophic interaction. In this case, however, only about 0.5 mmol of CH4 was produced. If this CH4 was produced solely from CO2 reduction by the electrons released from ethanol oxidation, 1.0 mmol of acetate should be produced which can be either used by the methanogen or accumulated in medium. The ethanol concentration for this treatment is slightly different from the earlier version, perhaps due to some mistakes. But in the absence of magnetite, it should be easy to calculate electron balance which shall help to better understand what is really going on. Second, based on Figure 1C, the treatment of Geobacter monoculture plus Fe3O4 indicated that a lot of ethanol (from 1.2 mmol decreased to about 0.4 mmol) was oxidized by Geobacter with Fe3+ in magnetite as the only electron acceptor. This indicates a substantial reduction of Fe3+ in magnetite. With this reduction activity involved, it is hard to image why DIET is still necessary for the living of Geobacter and if acetate is released by Geobacter during ferric iron reduction, DIET would be also unnecessary for the living of Methanosarcina.

---

## Round 0.3 · accepted · Accept

The authors have carefully addressed the comments of the reviewer and provided more discussion in the manuscript to improve the quality of the study.